# Robust and Fast Normal Mollification via Consistent Neighborhood Reconstruction for Unorganized Point Clouds

**DOI:** 10.3390/s23063292

**Published:** 2023-03-20

**Authors:** Guangshuai Liu, Xurui Li, Si Sun, Wenyu Yi

**Affiliations:** 1School of Mechanical Engineering, Southwest Jiaotong University, Chengdu 610031, China; 2Sichuan Province Informationization Application Support Software Engineering Technology Research Center, Chengdu 610103, China; 3Institute of Optics and Electronics, Chinese Academy of Sciences, Chengdu 610209, China; 4Sichuan Research and Design Institute of Agricultural Machinery, Chengdu 610066, China

**Keywords:** normal estimation, point cloud, feature preserving, normal mollification

## Abstract

This paper introduces a robust normal estimation method for point cloud data that can handle both smooth and sharp features. Our method is based on the inclusion of neighborhood recognition into the normal mollification process in the neighborhood of the current point: First, the point cloud surfaces are assigned normals via a normal estimator of robust location (NERL), which guarantees the reliability of the smooth region normals, and then a robust feature point recognition method is proposed to identify points around sharp features accurately. Furthermore, Gaussian maps and clustering are adopted for feature points to seek a rough isotropic neighborhood for the first-stage normal mollification. In order to further deal with non-uniform sampling or various complex scenes efficiently, the second-stage normal mollification based on residual is proposed. The proposed method was experimentally validated on synthetic and real-world datasets and compared to state-of-the-art methods.

## 1. Introduction

The normal information is a crucial geometric property of 3D point clouds, which is extensively applied in many fields, such as denoising [1,2], surface reconstruction [3,4], resampling [5] and consolidation [6], and feature detection and extraction [7]. The inaccurate normals are likely to result in the loss of detailed features on the point cloud surface, thereby negatively impacting the subsequent applications. Although many works have been proposed in this research field, normal estimation still confronts numerous challenges. First of all, normal estimation should have high precision, notably for noisy point clouds or non-uniformly sampled surfaces. Next, computational efficiency must be ensured to verify practical significance. Thirdly, adaptive parameters should be designed to apply the estimator to various conditions.

Regression-based normal estimation methods [3,8] are the most widely employed on account of their high efficiency and strong applicability. However, such methods are actually low-pass filters. Consequently, any sharp features of the surface will unavoidably be smoothed. In this case, normal mollification techniques [4,9] are proposed to improve the initial normal field. Nevertheless, these methods require refined parameter tuning and can no longer reliably recover curvature discontinuities under the excessive smoothing of sharp features. Additionally, segmentation-based approaches [10,11] are proposed to generate isotropic sub-neighborhood matching the current point to estimate normal. These methods can generate more reliable normals than other methods but usually require longer run times.

Motivated by the above consideration, we propose a fast, high-quality normal mollification method. In contrast with the conventional normal mollification process, a rough isotropic sub-neighborhood is sought and added in our method, which effectively avoids the requirement of reliable initial normals and refined parameters. Throughout the process, the normals of all points are initialized by a normal estimator of robust location. Then feature coefficient that measures the extent of points are near to sharp region is utilized to classify the point cloud into feature points and non-feature points. For feature points, an isotropic sub-neighborhood constructed by a Gaussian map and clustering is utilized for the first-stage normal mollification. Meanwhile, further secondary mollification can ensure steady and accurate estimation under various difficult scenarios. The experiments demonstrate that the presented method yields accurate and efficient results in the presence of noise and anisotropic samplings while preserving sharp features. Our main contributions can be summarized as follows:▪A normal estimator of robust location is used to estimate initial normals, which ensures the reliability of the smooth region normal.▪A robust recognition method based on normal differences is proposed to identify points close to sharp features. This method can accurately identify feature points at high noise levels.▪A robust normal mollification process based on neighborhood recognition is proposed, which can efficiently and reliably estimate normals for points around sharp features even in the presence of noise and non-uniform sampling.

## 2. Related Works

Normal estimation is challenging for point clouds with noise, non-uniformity of sampling, and sharp features. Considerable approaches have been developed for normal estimation in the literature, which are divided into the following six categories:

The first category is based on regression. The classical normal estimation method, proposed by Hoppe et al. [3] (PCA), defines the normal of a point as the eigenvector corresponding to the smallest eigenvalue of the covariance matrix of its neighbors, where the point’s local neighborhood is approximated by a plane. Mitra et al. [12] proposed adaptive neighborhood size to improve the robustness of the regression method to noise. However, the method tends to smooth sharp features and thus cannot correctly estimate the normal near the edges. Moreover, by assigning Gaussian weight to the current point’s neighbors during the plane fitting, a weighted version PCA is proposed in [8,13], which benefits from weakening the influence of some points, such as noise and outlier points. To better adapt to the shape of the underlying surface, Cazals et al. [14] and Guennebaud [15] utilize higher-order quadric surfaces and algebraic spheres to replace the plane. Mederos et al. [16] extend the plane fitting by introducing a robust statistic approach called M-estimator to reduce the impact of the neighbors belonging to different surface patches, but Newton’s method is needed to solve the normal; thus, the calculation is relatively complicated. Wang et al. [17] further consider the normals’ likeliness to penalize neighbors having normals much different from the normal of the current point. Recently, Sanchez et al. [18] introduced an iterative weighted PCA (IterWPCA) from a robust M-estimator and effectively dealt with the noise and anisotropy problems. However, the specified curvature and noise scale assigned to the global may cause the algorithm not to converge to the optimal solution.

Another train of thought is based on the Voronoi diagram, which was first proposed by Amenta [19]. In this method, the furthest pole of the Voronoi lattice is used to approximate the normals, but it works only for the noise-free point clouds. Dey et al. [20] extend the idea by seeking the Delaunay sphere and refining the poles, thereby being able to estimate the normal while the presence of noise. Based on the former theories, Alliez et al. [21] present a method that combines the advantages of PCA and Voronoi-based to achieve more stable normal estimation results.

To lessen the impact of noise and outlier points while improving the initial normals, the methods based on normal mollification are studied. Yagou et al. [22,23] propose filters to mollify the normal field locally and introduce three kinds of filters: mean filter, median filter, and alpha-trimming filter to reduce the sensitivity to noise while recovering curvature discontinuities. Similarly, Jones et al. [9] extend bilateral filtering to estimate normals. Öztireli et al. [4] expound the implicit least square surface from the perspective of local kernel regression, and take the robust local nuclear regression method in robust statistics for reference, proposing a robust normal mollification method. Additionally, a half-quadratic regularization method [24] also restores sharp features while improving noisy normals. Although normal mollification methods can obtain virtually correct normal [25] for the points close to sharp features, as post-processing methods, all of them require dependable initial normal.

The methods mentioned above are limited in the capability to maintain sharp features; hence the fourth category based on the voting method is developed. Li et al. [26] propose a robust local noise estimation method combined with kernel density estimation, which votes all the neighborhood points on a plane set determined by arbitrary triples to select the optimal tangent plane. This approach can accurately estimate the normal of feature points and have strong robustness to noise and outliers. However, it does not take non-uniform sampling into account. To handle this issue, a uniform sampling technique based on the randomized Hough transform is proposed by Boulch and Marlet [27]. Whereas in the case of a large dihedral angle, the difference in normals of plane sampling will be trivial, so these normals will vote for the identical bin, resulting in the blurring of the normals near its edge. Additionally, Zhang et al. [25] introduce a pair consistency voting algorithm (PCV) to gain the optimal tangent plane via point pairs voting between neighborhood points and use density weights to solve non-uniform sampling issues. In order to deal effectively with noise and outliers, Mura et al. [28] proposed a method based on robust statistics that can simultaneously maintain the sharp features of the point cloud. Such methods usually consider that points with large fitting residuals have a negative impact on the normal estimation and should be reduced or dropped during the plane fitting.

The fifth category is based on neighborhood segmentation. Fleishman et al. [10] propose robust moving least squares to segment the neighborhood into multiple piecewise smooth regions, whereas normal estimation of this method is on the premise of reconstruction; thus, it is time-consuming. Zhang et al. [11] design an unsupervised learning process, adopting the low-rank subspace clustering method (LRR) with prior knowledge to divide into multiple isotropic neighborhoods. However, a lot of time is taken to solve the model in each segmentation process. To reduce the computational time, based on LRR technology, Liu et al. [29] utilize the least square method as a guide to segmentation neighborhood, decreasing time and ensuring its high-quality segmentation. Moreover, Yu et al. [30] proposed a neighborhood segmentation-based and neighborhood growth-based method to construct a consistent neighborhood with the current point, but this method could be labile while dealing with sparse sampling models. Different from pure segmentation, Cao et al. [31] obtain the surface patch consistent with the current point through neighborhood shift techniques to estimate accurate normals.

Recently, the sixth category of learning-based normal estimation methods has been proposed [32,33,34,35]. Zhou et al. [36] offer a multi-scale neighborhood selection technique and an additional plane feature constraint based on PCPNet [32] to enhance performance. By leveraging both the PointNet and 3DCNN, Hashimoto et al. [37] proposed a joint network that can accurately infer normal vectors from a point cloud. Zhou et al. [38] proposed a normal filter based on multipatch stitching. Thanks to their patch-level architecture, their method can reduce computational costs and improve the robustness of noise removal. Boulch et al. [39] proposed to convert a local point cloud block into a 2D Hough space accumulator by randomly selecting a point triplet and voting for the normals in that plane. Then, the normals are estimated from the accumulator as a continuous estimate of the regression problem. This method does not take full advantage of the 3D information, as it loses information in the transformation phase. Later, a mixture-of-experts (MOE) architecture called Nesti-Net was introduced by Ben-Shabat et al. [40] and relies on a data-driven methodology to determine the ideal scale around each point and boosts sub-network specialization. Usually, such methods are suitable to models with many curvature details and/or high noises, and many methods are still limited in their ability to handle piecewise smooth surfaces and preserve sharp features.

## 3. Overview

Given a point cloud P={pi}i=1n as input, our method takes three steps to estimate the normals. (1) The nearest neighborhood N with S neighbors is obtained for each point pi, and initial normals are referenced for NERL. Afterward, point cloud P is classified into two types: feature point set Pf or non-feature point set Pn, according to feature coefficients of the points, which are detailed in Section 4. (2) Different neighborhood N˜ is selected for each feature point. After that, Gaussian map and clustering are utilized to form different clusters {N¯1,N¯2,...,N¯K}, where K is the number of clusters of the current point. Each cluster is solved to a plane by the RANSAC algorithm. The residuals between the current point and each fitting plane are calculated, then the cluster that matches the current point is regarded as the optimal cluster that is employed to conduct the normal mollification, as shown in Section 5.1. (3) According to the distribution of fitting residuals, the feature points with larger residuals can be considered abnormal points. Then the second-stage normal mollification is carried out for these points, which is explained in Section 5.2. The overview of our method is concluded in Algorithm 1 and illustrated in Figure 1.
**Algorithm 1 The pipeline of our algorithm****Input:** Point cloud P;**Output:** Normal set {ni};1: **for** pi∈P **do**2:  Compute initial normal n˜i∈pi, feature coefficient wi;3:   **if** wi<wt **then**4:    ni=n˜i;5:   **else**6:    Obtain mollification neighborhood N¯i∈pi via Algorithm 2;7:    Obtain ni via the first-stage normal mollification based on Equation (7);8:     **if** fitting residual r(pi)>εf **then**9:      Obtain mollification neighborhood Ni∈pi via Section 5.2;10:       Obtain ni via the second-stage normal mollification based on Equation (8);11: **end for**

## 4. Normal Initialization and Feature Point Recognition

In order to make the initial normals of smooth areas have a certain accuracy, as well as preparation for the subsequent processing, based on the weighted principal component analysis (WPCA) [13], our NERL takes advantage of M-estimates of location [41] instead of the mean to estimate the normals. For each point $p_{i}$ and its S neighbors, a covariance matrix C is calculated as
(1)$$\begin{array}{l} C=\frac{1}{\sum w_{d}(p_{j},p_{i})}\sum_{j=1}^{S}w_{d}(p_{j},p_{i})\cdot (p_{j}-p_{r}){\left(p_{j}-p_{r}\right)}^{T}\\ p_{r}=\frac{\sum_{j=1}^{S}w_{d}(p_{j},p_{i})\cdot p_{j}}{\sum_{j=1}^{S}w_{d}(p_{j},p_{i})}, \end{array}$$
where $w_{d}$ and $p_{r}$ are a distance weight and points of robust location, respectively, and $w_{d}(p_{j},p_{i})=\exp(-||p_{j}-p_{i}|{|}^{2}/2\cdot \sigma_{d}{}^{2})$. $\sigma_{d}$ is set to the average distance from its ten nearest neighbors of the current point to their one respective neighbor. The estimated normal at $p_{i}$ is the eigenvector associated with the smallest eigenvalue of the covariance matrix. Figure 2 shows the normal estimation errors of PCA, WPCA, and NERL on the Harpago and Scallop curved surface models.

The normals of the points in the smooth regions are reliably computed through NERL, whereas those of the points near the edges still fail to preserve sharp features. Therefore, it is indispensable to distinguish the feature points further. In order to make feature detection more applicable, we divided it into two cases according to the noise scale to deal with separately. Under the condition of small-scale noise, for each point $p_{i}$, the eigenvalues of its covariance matrix computed by PCA and NERL methods can be obtained by singular value decomposition, and the feature coefficients $w_{i1},w_{i2}$ are calculated by their three eigenvalues, respectively, corresponding to $\left(\lambda_{0},\lambda_{1},\lambda_{2}),({\lambda^{\prime }}_{0},{\lambda^{\prime }}_{1},{\lambda^{\prime }}_{2}\right)$ with ascending order.
(2)$$w_{i1}=\frac{\lambda_{0}}{\lambda_{0}+\lambda_{1}+\lambda_{2}},w_{i2}=\frac{{\lambda^{\prime }}_{0}}{{\lambda^{\prime }}_{0}+{\lambda^{\prime }}_{1}+{\lambda^{\prime }}_{2}}.$$

Feature points $P_{f}$ are differentiated via the given threshold $w_{t1},w_{t2}$, i.e., $P_{f}=\{p_{i}\in P|(w_{i1}>w_{t1})\cup (w_{i2}>w_{t2})\}$. Although the above method has good efficiency, it will become invalid under the situation of large-scale noise, as illustrated by Figure 3. Accordingly, a robust method based on normal differences is proposed. For a given point $p_{i}$ with the neighborhood $N_{p_{i}}$ of S neighbors, a covariance matrix U is constructed by the normal difference of point pairs in the neighborhood:(3)$$U=\frac{1}{S}\sum_{n_{j},n_{k}\in N_{p_{i}}}(n_{j}-n_{k}){\left(n_{j}-n_{k}\right)}^{T},$$
where $n_{j}$ and $n_{k}$ are the normals of the neighborhood points. The size of the three eigenvalues of the covariance matrix U reflects the dispersion degree of the normals in the neighborhood. For smooth regions, the three eigenvalues tend to be zero, and as the enlargement of dispersion degree, the size of the eigenvalues increases accordingly. Thus, for a point $p_{i}$, its feature coefficient $w_{i3}$ is defined as the sum of three eigenvalues. We still specify a threshold $w_{t3}$, and feature points are denoted as $P_{f}=\{p_{i}\in P|(w_{i3}>w_{t3}\}$. Figure 3 indicates that our method can still perform accurate feature detection even if under large-scale noise. Note that in practical implementation, there is no strict standard for the selection of both, which depends on the prior cognition of the processing object.

## 5. Robust Normal Estimation

In this section, we explain how to reliably estimate the normals of the distinguished feature points via the normal mollification methodology, which divides into two successive steps: a preliminary normal estimation based on neighborhood segmentation and a further normal estimation based on residual under various difficult scenarios.

### 5.1. Normal Mollification Based on Neighborhood Segmentation

**Gaussian map and clustering:** The Gaussian map of the point cloud is used to map the normals of the surface in Euclidean space to the unit sphere. Based on the fact that the normals of points change continuously in the smooth regions but alter dramatically when located at the sharp surface, the normals of smooth areas will form a single cluster, while those near the sharp feature will form multiple clusters on the Gaussian sphere and each cluster corresponds to different surfaces, as shown in Figure 4. Since the fitting-based approaches generally assume that the surface is smooth everywhere and the estimated normals are continuous at sharp edges, which makes the threshold employed for subsequent clusters unreliable, we temporarily remove the feature points from the neighborhood in the first-stage normal mollification to form a single cluster for each patch on the Gaussian sphere. Therefore, for each feature point $p_{i}\in P_{f}$, neighborhood $\tilde{N}$ of $\tilde{S}$ neighbors without feature points will be selected.

The dataset of the neighborhood of the feature point $\tilde{N}=\{p_{i},i=1,2...,\tilde{S}\}$ is mapped to the Gaussian sphere, and the mapped dataset can be expressed as $G(\tilde{N})=\{n_{i},i=1,2...,\tilde{S}\}$, where $n_{i}$ is the corresponding normal. Since the amount of sub-neighborhood is not foreknown, the parameter-free clustering algorithm is adopted. Thus, here, we select the parameterless hierarchical agglomerative clustering (HAC), where the bottom-up clustering procedure is also appropriate for this process. In this algorithm, each data point is regarded as a separate cluster beforehand. Then, according to the similarity between clusters, they are gradually merged into increasing clusters until the similarity between any two clusters exceeds a specified threshold.

Calculating the similarity between two clusters, ${\tilde{N}}_{1}$ and ${\tilde{N}}_{2}$, is also termed the linkage criterion, which includes three categories: single linkage, complete linkage, and average linkage. The common average linkage shows the best performance in our application:(4)$$D_{ave}({\tilde{N}}_{1},{\tilde{N}}_{2})=\frac{1}{\left|{\tilde{N}}_{1}\right|\cdot \left|{\tilde{N}}_{2}\right|}\sum_{i\in {\tilde{N}}_{1}}\sum_{j\in {\tilde{N}}_{2}}d(i,j),$$
where $\left|{\tilde{N}}_{1}\right|,\;\;\;\left|{\tilde{N}}_{2}\right|$ represent the number of points in clusters ${\tilde{N}}_{1},{\tilde{N}}_{2}$, respectively; i and j denote i-th and j-th points in the clusters ${\tilde{N}}_{1},{\tilde{N}}_{2}$; d(i,j) is expressed as the similarity between two points; and
(5)$$d(i,j)=\arccos(n_{i}\cdot n_{j}),$$
where $n_{i}\cdot n_{j}$ is the inner product of two unit normals. On the Gaussian sphere, they will be merged when the similarity between two clusters is less than a given threshold δ.

An appropriate threshold δ is key to obtaining credible clustering results. A small threshold can result in the isotropic neighborhood being divided into multiple clusters. Inversely, a relatively large threshold makes the anisotropic neighborhood group into an identical cluster. However, on account of the feature points being removed in advance, the normals of different patches have obvious recognizability; thus, the appropriate threshold is in a relatively large tolerance. After considerable tests, the threshold of $7^{\circ }$ performs good outcomes in this algorithm.

**Neighborhood recognition:** The dataset in the neighborhood of feature points N˜={pi,i=1,2...,S˜} is grouped into several isotropic clusters {N¯1,N¯2,...,N¯K} by using a Gaussian map and clustering, where K is the number of clusters. To eliminate the influence of noise and outliers, the RANSAC algorithm is utilized for each cluster. Then the residuals between the current point and each fitting plane are obtained, and the cluster corresponding to the minimum residual is selected as the mollification neighborhood of the current point. The cluster and neighborhood recognition algorithms are shown in Algorithm 2.
**Algorithm 2 Neighborhood recognition of feature points****Input:** pi,pi∈ PF;**Output:** pi‘s mollification neighborhood N¯i;1: **for** i=0 to PF.size() **do**2:   Seek S˜ nearest neighborhood N˜;3:   View every point p1,p2,...pS˜ as a single cluster {N˜1,N˜2,...,N˜S˜};   //see Section 5.1 and Figure 44:   **for**
j=1 to S˜
**do**5:     Compute similarity: Dave(N˜p,N˜q),   p≠q based on Equation (4);6:     T=min{Dave(N˜p,N˜q)}; 7:     **if** (T<δ) **then**8:      Find two cluster N˜m,   N˜n with minimum similarity;9:      Merge (N˜m,N˜n);10:     **else**11:      Obtain the ultimate cluster result: {N¯1,N¯2,...,N¯K};12:      **break**;13:   **end For**14:   {ε1,ε2,...,εS}=computeResidual (pi,{N¯1,N¯2,...,N¯S}); //see Section 5.215:   N¯i=min{ε1,ε2,...,εS}corresponding to N¯c, c∈{1,2,...,S};16: **end for**17: **return** N¯i with S¯i neighbors;

**Mollification:** The isotropic sub-neighborhood matching the current point is employed to mollify its normal. Similar to previous studies [5], our iteration formula of the first-stage normal mollification is given by Equation (6).
(6)$$n_{i}^{k+1}=\frac{\sum_{j=1}^{{\bar{S}}_{i}}w_{n}(n_{i}^{k},n_{j})n_{j}}{\sum_{j=1}^{{\bar{S}}_{i}}w_{n}(n_{i}^{k},n_{j})},$$
where $n_{j}$ is the normal of a point in the mollification neighborhood ${\bar{N}}_{i}$ of the current point $p_{i}$, $n_{i}^{k}$ is normal of $p_{i}$ after the k-th mollification, and $w_{n}(n_{i}^{k},n_{j})$ is a normal deviation kernel function as shown in Equation (7).

As the deviation between the normal of the k-th iteration and the normal of the (k−1)-th iteration is less than the threshold ε, the iteration will be terminated, i.e., $1-n_{i}^{k-1}\cdot n_{i}^{k}\leq \varepsilon$, and the threshold $\varepsilon =10^{-4}$ is set.
(7)$$w_{n}(n_{i}^{k},n_{j})=\exp(\frac{-||n_{i}^{k}-n_{j}|{|}^{2}}{\sigma_{n}^{2}}),$$
where ||⋅|| is the $l_{2}-norm$, and $\sigma_{n}$ is the normal deviation bandwidth. To make the normal mollification more accurate, the selection of this parameter should reduce the influence of outlier normals as much as possible in the normal mollification process. In the practical utilization, $\sigma_{n}$ is set to the maximum normal deviation, i.e., $\sigma_{n}=\max(\left\|n_{i}-n_{j}\right\|),\;\;j\in [1,{\bar{S}}_{i}]$.

### 5.2. Normal Mollification Based on Residual

**Detection and neighborhood recognition of abnormal points:** Due to the existence of noise, non-uniform sampling, or various complications, some non-feature points are over-identified as feature points, such as sharp corners, narrow-bands, and low-sampling density regions, so the normals in some points may be incorrectly estimated.

To cope with this kind of issue, as well as compensate for the impact of excluded feature points in the first stage, the second-stage normal mollification is devised. First of all, the residual set $\left\{r(P_{f})\right\}$ between each feature point and the consistent optimal plane can be calculated, and abnormal points $P_{a}$ are defined as $P_{a}=\{\left.p_{i}\in P_{f}\right|r(p_{i})>\varepsilon_{f}\},\varepsilon_{f}=\bar{\mu }+3\sigma$, where $\bar{\mu }$ and σ are the mean and standard deviation of the residual set, respectively. Note that it is impossible for the recognition to be 100% precise. However, excessive identification only increases the running time of the method and does not degrade the results.

Next, similar to the first stage, we need to seek the mollification neighborhood of the abnormal points for normal readjustment. Therefore, a neighborhood with a larger size $S^{*}=6\tilde{S}$ is selected. Since the normals of most feature points are correctly estimated in the first stage, they would be involved when constructing the second-stage mollification neighborhood. Given an abnormal point $p_{i}$ and its neighbor $p_{ij}$, we define the residual $r_{ij}=\left|n_{ij}\cdot (p_{ij}-p_{i})\right|$. The mollification neighborhood $N_{i}$ is constituted by selecting neighbors if $r_{ij}$ is less than the specified threshold $\sigma_{\tau }$, which is set to 0.25 degrees, corresponding to the amplitude in this article.

**Mollification:** After the abnormal points’ mollification neighborhood is sought, the second-stage normal mollification is designed to estimate the normals of these points more finely. The formula of the second-stage normal mollification is given by Equation (8).
(8)$$n_{i}=\frac{\sum_{p_{j}\in N_{i}}w_{s}(r_{ij})n_{ij}}{\sum_{p_{j}\in N_{i}}w_{s}(r_{ij})},$$
where $w_{s}(r_{ij})$ is a residual kernel function defined as $w_{s}(r_{ij})=\exp(-{\left(r_{ij}/\sigma_{s}\right)}^{2})$, and $\sigma_{s}$ is set as $\sigma_{\tau }/3$.

## 6. Implementation Results

To evaluate the performance of the proposed method, a variety of point cloud models of sharp features with a complicated neighborhood, non-uniform sampling, and noises are tested. We contrast our method with some state-of-the-art approaches: PCA [3], BF [4], PCPNet [32], DeepFit [33], MTRNet [34], RNE [26], HF [27] (only HF_cubes is involved), IterWPCA [18], and PCV [25]. The learning-based methods used for comparison are trained via the PCPNet shape dataset [30]. For the sake of fair comparisons, the same neighborhood size is applied to each algorithm, and all other parameters required by other methods, if any, are set to the default or the values of the best results obtained through trial and error. In addition, the parameters of our algorithm are summarized below:

$S,\tilde{S},S^{*}$ are the number of neighbors for calculating the initial normal, the first-stage, and the second-stage mollification, respectively ($\tilde{S}=2S,S^{*}=6\tilde{S}$).
$w_{t1},w_{t2}$ are the thresholds used to distinguish feature points under low-level noise.$w_{t3}$ is the threshold used to distinguish feature points under high-level noise.δ is the threshold for clustering ($\delta =7^{\circ }$).$\sigma_{n},\sigma_{s}$ are the normal deviation and the residual bandwidth for the first-stage and the second-stage mollification, respectively.

In this paper, three different scores are used to evaluate the performance of the algorithms quantitatively: (1) the runtime; (2) the root mean square with a threshold ($RMS_{\tau }$); and (3) the number of bad points (NBPs), where (2) and (3) are defined as
(9)$$RMS_{\tau }=\sqrt{\frac{1}{\left|P\right|}\sum_{p\in P}f{\left(\hat{n_{p},{\tilde{n}}_{p}}\right)}^{2}},$$
where
(10)$$f(\hat{n_{p},{\tilde{n}}_{p}})=\left\{\begin{array}{c} \hat{n_{p}{\tilde{n}}_{p}},if\;\;\;\hat{n_{p}{\tilde{n}}_{p}}<\tau \\ \pi /2,\;\;\;otherwise \end{array}\right.,$$

$n_{p}$ is the ground-truth normal of p, and ${\tilde{n}}_{p}$ is the estimated normal of p. $\hat{n_{p}{\tilde{n}}_{p}}$ is the angle between $n_{p}$ and ${\tilde{n}}_{p}$. $\tau =10^{\circ }$ is set, which means those points whose errors are more than $10^{\circ }$ are regarded as bad points.

All the noise used in the experiments is Gaussian noise, with different standard deviations as a percentage of the bounding box diagonal. Our method is implemented using Microsoft Visual Studio 2015 with C++ and Point Cloud Library (PCL) [42]. In the specific test process of our algorithm, the parameters $\tilde{S}=2S$, $w_{t1}=0.02$, $w_{t2}=0.012$, and $w_{t3}=0.2$ (note that the neighborhood size below refers to $\tilde{S}$) are the default unless otherwise specified. All the experiments have been performed on the same computer with Inter(R) Core(TM) i3-7100 3.90 GHZ and 16 GB RAM without parallel computing.

### 6.1. Quantitative Evaluation

#### 6.1.1. Accuracy

To evaluate the effectiveness of our method under various complex conditions, three basic categories are used for performance testing, as shown in Figure 5. Here, the first category is used to evaluate performance on sharp features, the second is a test of robustness to non-uniform samplings, and the third is utilized to assess the ability to preserve small structures. For each model, the neighborhood size is defined as 40 points. The second recognition strategy only is used in Detials in our method. Additionally, for PCPNet and MTRNet, we adopt the multi-scale versions provided by authors with the default parameters. DeepFit does not have a multi-scale version, so the 256 neighborhood points and 3-order jet recommended by the author are adopted. The results for each model are presented in Figure 6 (the errors of Pipe and Icosahedron are the average of the test results of gradient and striped anisotropic samplings).

Overall, PCA generates overly smooth normals near sharp edges, which can be markedly seen from the NBPs of the first two categories of models in Figure 7, but it has a slight improvement in the processing of more curved models. PCPNet obtains the worst result due to its poor accuracy on smooth regions. Furthermore, DeepFit, as a deep-fitting method, underperforms in the face of sharp features. In contrast, MTRNet obtains better results, especially for detecting details. Although BF and IterWPCA maintain sharp features to a certain extent, there are still considerable bad points, as presented in Figure 7. HF and RNE have almost similar results, and RNE is inferior to HF in the Details module, whereas it has some advantages in anisotropic sampling. PCV achieves good outcomes in handling the above cases, but in comparison, our method takes on lower and lower computational costs (see Section 6.3).

#### 6.1.2. Robustness to Noise

In this experiment, two models of Cube and Lagera with variational Gaussian noise levels are tested, where the Cube model focuses on relatively small noise levels varying from 0% to 1.5% with an interval of 0.25% and the second recognition strategy is used when the noise level reaches 0.5%; the Lagera noise data from the PCPNet dataset centers on relatively large noise levels, which contains noise varying from 0% to 10% with an interval of 2%, so the second recognition strategy is used, and $w_{t3}$ is set as 0.6 when the noise level exceeds 6%. The neighborhood size is defined by 60 points for each model. The statistical results and the visualization of the angular error are presented in Figure 8 and Figure 9, respectively. Compared with other methods, due to noise affecting the stability of voting results, HF has bad behavior under different noises of the two models. The accuracy of the normal estimated by MTRNet in the Lagera model containing curved surfaces is well guaranteed with the increase in noise level, whereas on account of incorrect estimation of normals at sharp features and noisy normal, as shown in Figure 9, it has a high $RMS_{\tau }$ in the Cube model. RNE has great contrast between models with different features. In comparison, it is bad at large noises and curved models. The errors of IterWPCA and PCV have moderate results but a great amplitude rise with the increase in noise level. The proposed method achieves the lowest $RMS_{\tau }$, which indicates that our method is more advantageous in robustness to noise.

### 6.2. Qualitative Evaluation

We further demonstrate the performance of our method in point cloud filtering applications. In the experiment phase, we utilize an effective point updating algorithm [1,43], a point cloud smoothing algorithm Bilateral method [44], and EGT [45], in which the normal information used in this algorithm is matched by the estimated normal output via each method. To obtain better results in the experiment, we alternately call the normal estimation and position update for several iterations. Because the excessive implementation of this algorithm could smooth out sharp edges, the number of iterations is empirically set to 4. The Joint model with 48K points and 1.0% noise is used in our experiment, where except for the first iteration, the first recognition strategy is used in the remaining iterations, and the results are shown in Figure 10. We can see that our method has higher fidelity in preserving features than other methods.

Real scan data often has many defects, such as noise, outliers, and non-uniform sampling. In order to illustrate the effectiveness of our method on real scan data, we tested our method on the Pyramid and Office models and compared them with PCA, as shown in Figure 11, where the first recognition strategy is adopted in our method. As we can see, the PCA method results in the loss of the sharp edges and corners, but the sharp features are favorably preserved by our method.

### 6.3. Computational Time

We present the time consumption of all algorithms applied to the models in Figure 5 in Table 1. It can be seen that PCA and BF are the fastest two methods, whereas BF has a larger improvement in error results. Due to the voting strategy of point pairs, PCV is the slowest method, but it has a better effect. Moreover, learning-based approaches take relatively more time. Our method is slightly faster than RNE and HF, and the accuracy has been significantly improved. Moreover, it is worth noting that the proposed method can be fully parallelized easily, which will remarkably reduce the computational time taken by our method.

## 7. Conclusions

In this article, we present a robust normal estimation method for point clouds with sharp features, where via a normal estimator of robust location and a robust feature point recognition, reliable initial normals of smooth regions and accurate recognition of points close to sharp regions can be obtained. Further, the two-stage normal mollification based on neighborhood recognition ensures our algorithm can deal with various difficult conditions. We demonstrate the validity of our work by testing on both synthetic and scanned point clouds. Compared with other algorithms, the proposed method has higher quality, lower computational cost, and robustness to noise and non-uniform sampling. Moreover, it is simple and easy to implement.

However, more faithful normals can be generated with delicate parameters turning. In the future, we would like to choose these parameters adaptively according to various noises and sampling densities.

## Figures and Tables

**Figure 1 sensors-23-03292-f001:**
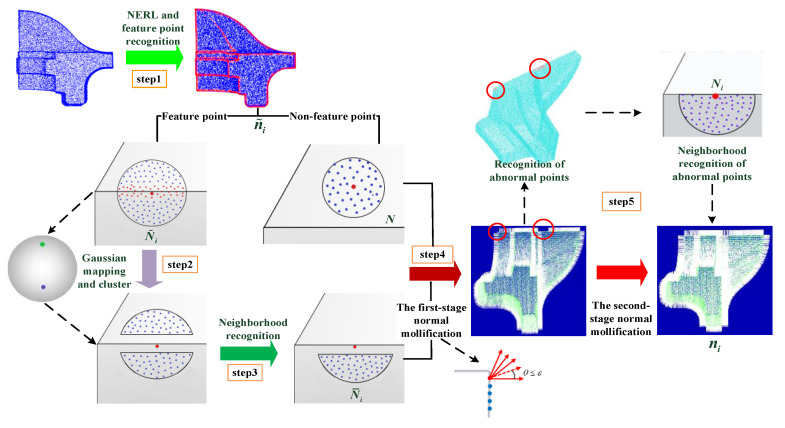
The pipeline of our algorithm (***N*** represents the neighborhood of each stage, and ***n*** denotes the normal of the stage).

**Figure 2 sensors-23-03292-f002:**
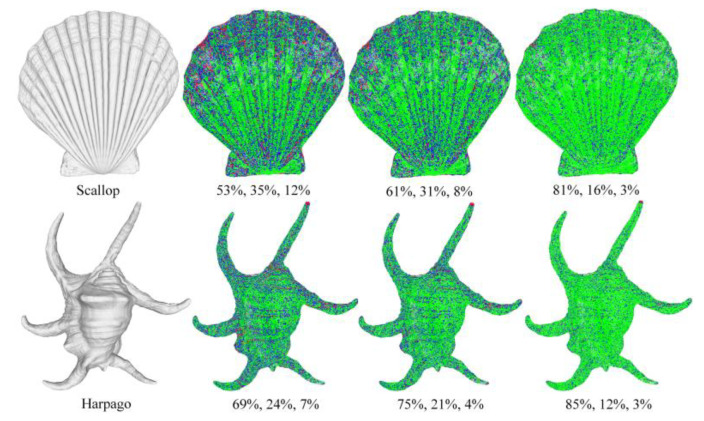
The normal estimation error. Green denotes points with an estimated normal that deviates by less than 5° from the ground-truth normal; blue encodes angular deviations between 5° and 10°; and red marks errors more than 10°. From left to right are the model, PCA, WPCA, and NERL methods, and the percentages of green, blue, and red points are counted under each result.

**Figure 3 sensors-23-03292-f003:**
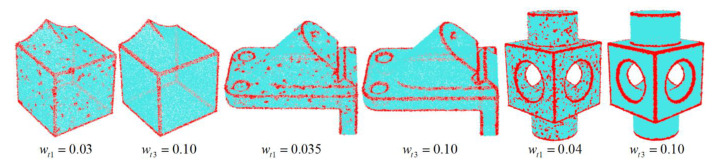
Feature detection results by common method [11] and our method under large-scale noise. Left: smooth-feature model with 0.25% Gaussian noise. Middle: anchor model with 0.30% Gaussian noise. Right: block model with 0.30% Gaussian noise.

**Figure 4 sensors-23-03292-f004:**
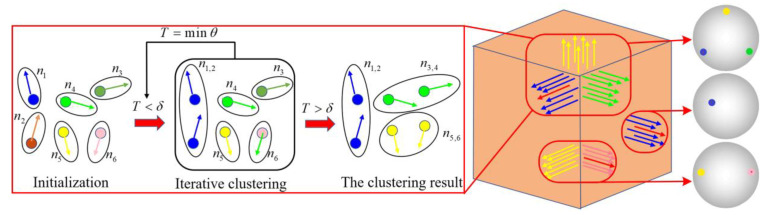
Gaussian map and clustering of cube.

**Figure 5 sensors-23-03292-f005:**
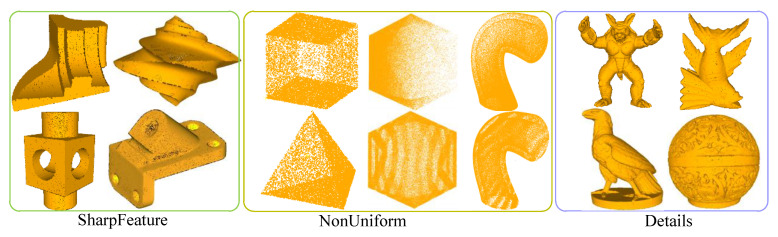
The point cloud models used to test the experiment.

**Figure 6 sensors-23-03292-f006:**
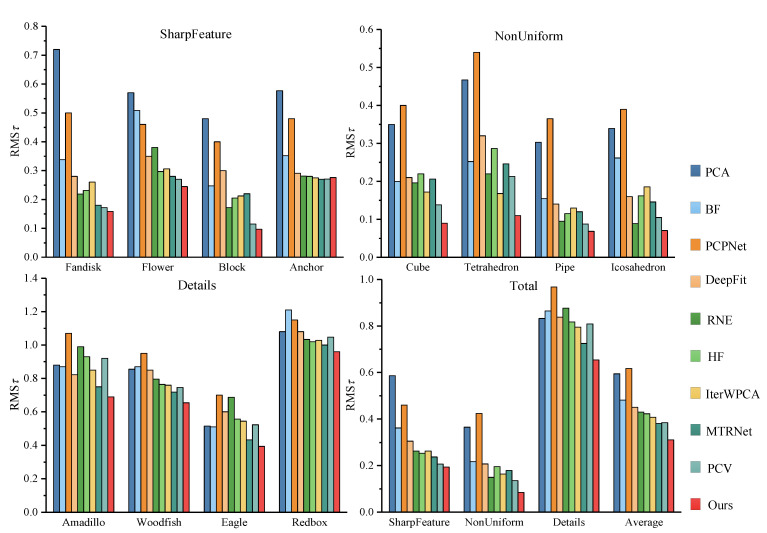
The ${\mathrm{RMS}}_{\tau }$ estimated by various methods, where the first three respectively represent the angular error of sharp feature models, non-uniform sampling models, and detail models; the comparison of the comprehensive performance of each algorithm in each category is demonstrated at the end, in which the average errors of each category and all models are included.

**Figure 7 sensors-23-03292-f007:**
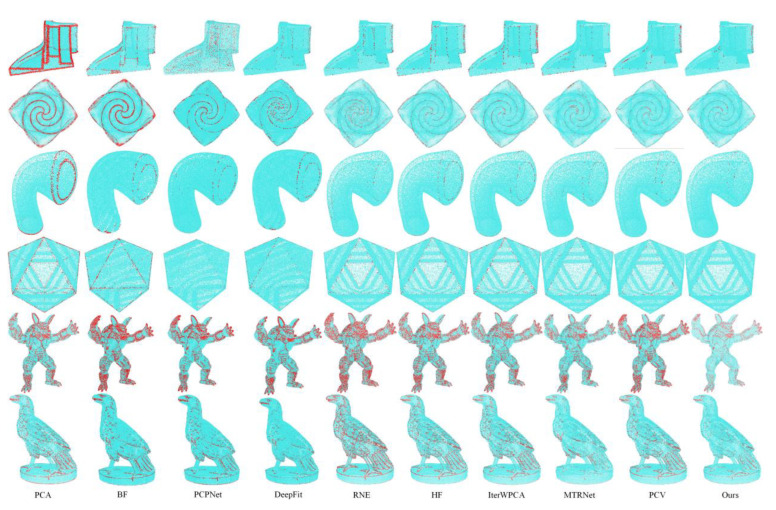
Visualization of NBPs on partial models of SharpFeature, NonUniform, and Details.

**Figure 8 sensors-23-03292-f008:**
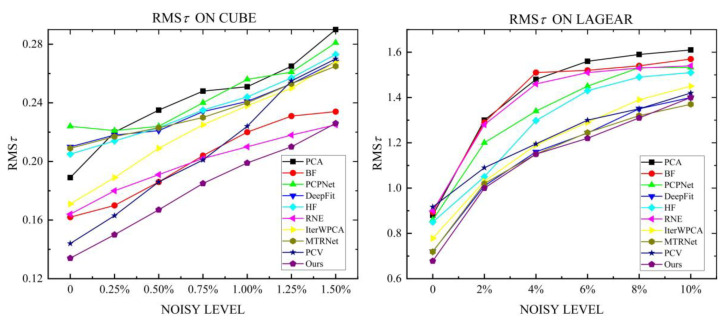
Comparison of the robustness of different algorithms to noise on Cube models with small noise levels and Lagera models with large noise levels.

**Figure 9 sensors-23-03292-f009:**
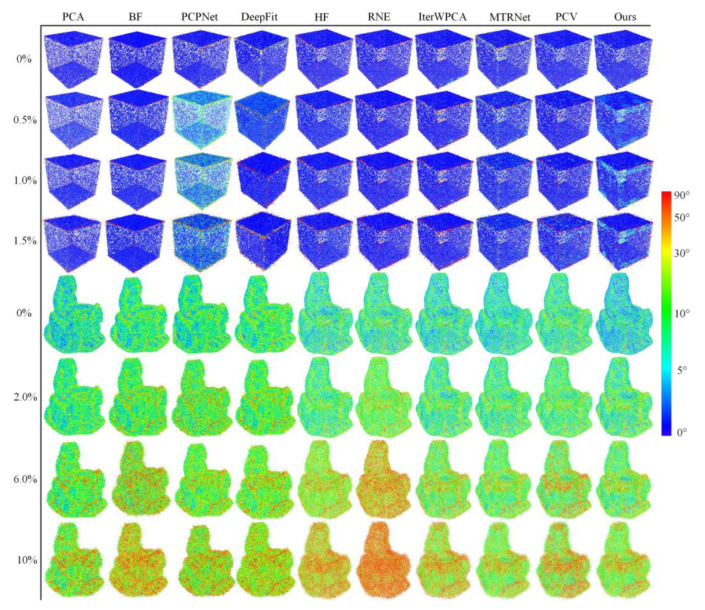
Visualization of angular error evaluated in Cube and Lagera models in the experiment of robustness to noise.

**Figure 10 sensors-23-03292-f010:**
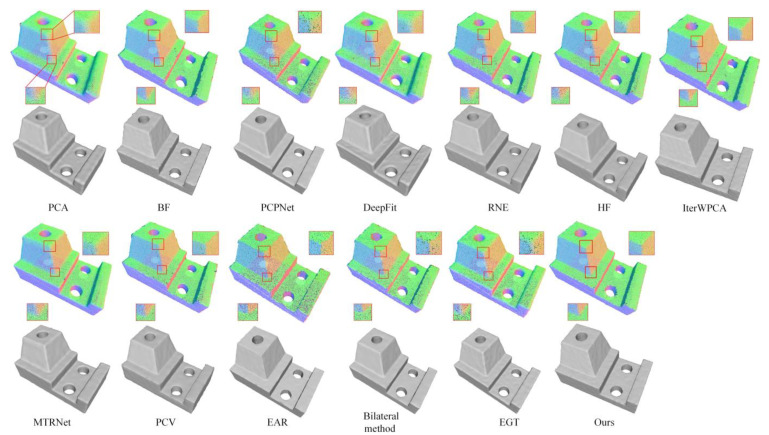
Comparison on point cloud filtering. First row: filtering results with upsampling. Second row: corresponding to surface reconstruction results.

**Figure 11 sensors-23-03292-f011:**
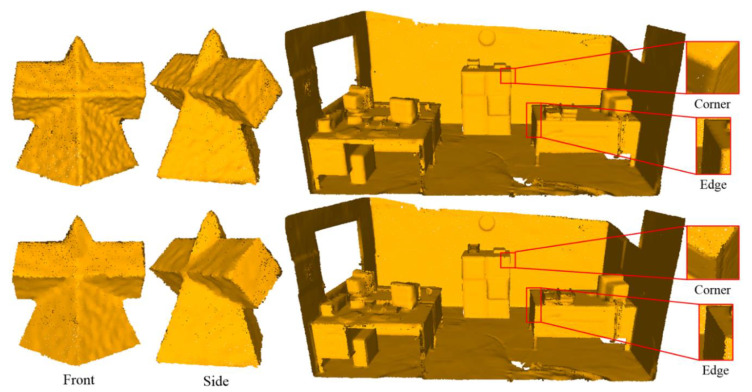
The normal estimation results on real scan data. The result of PCA is shown in the top row, and the result of the proposed method is shown in the bottom row.

**Table 1 sensors-23-03292-t001:** Mean computational time (in seconds) of different algorithms.

	PCA	BF	PCPNet	DeepFit	MTRNet	RNE	HF	IterW -PCA	PCV	Ours
SharpFeature (48,068)	0.07	0.56	61	59.7	42.5	23.2	15.7	6.25	184.7	2.98
NonUniform (72,000)	0.63	0.98	90.8	78.3	64.5	42.55	17	14.1	296.7	4.34
Details (189,021)	1.43	3.82	249.5	208.2	161.6	84.5	76	22.3	523.7	32.5
Average	0.71	1.79	133.8	115.4	89.5	50.1	36.2	21.3	335	13.3

## Data Availability

Not applicable.

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
