# Peer review of "Robust and Fast Normal Mollification via Consistent Neighborhood Reconstruction for Unorganized Point Clouds"

_sensors, 2023, doi:10.3390/s23063292_

Round 1
Reviewer 1 Report
In this paper a normal mollification approach is presented. The first step is to extract the normals with a standard (weighted) PCA-based approach on a given neighborhood per point. The second step is to identify the feature points based on the eigenvalues of the PCA and also on the PCA applied on the computed normals for each neighborhood.
The non-feature points are assigned the normals of the PCA-based approach while on the feature points the neighborhoods are further refined with Gaussian mapping to belong solely on smooth regions and not cross edge features. Then on these neighborhoods an iterative mollification procedure is applied to refine the normals of the feature points.
Major comments.
1. Please give a formal definition of what is the correct normal of a feature point. As you are aware there exist a lot of definitions of what a proper normal for a feature point is.
2. Several papers use also PCA on the normal information as in equation (3).
3. Note that equations (6, 8) are similar in nature to equation (2) of ref.[5]. So mention this and explain the differences of your approach with their approach. In [5] they use this equation to separate feature from non-feature points. Explain why your approach is better than their approach.
4. In the experiments please use all the methods in the figures not some on one experiment and some in other. All the approaches need to be present.
5. Point cloud smoothing is a straightforward application of the presented approach, yet the experiments shown are not very convincing. This method has to be compared against [5] and Alexander Agathos, Philip Azariadis, Sofia Kyratzi, Elliptic Gabriel Taubin smoothing of point clouds, Computers & Graphics, Volume 106, 2022, Pages 20-32, https://doi.org/10.1016/j.cag.2022.05.009.
Also, the work of Shachar Fleishman, Iddo Drori, Daniel Cohen-Or, Bilateral mesh denoising, ACM Transactions on Graphics, Volume 22, Issue 3, July 2003 pp 950–953, https://doi.org/10.1145/882262.882368
uses a procedure for obtaining normals that is directly applicable to point clouds and can be considered also in this work.
6. Line 255: “robust RANSAC algorithm”: what is this algorithm, it is different than RANSAC (which is not performing well with noisy data)?
7. Real-world scan data can easily reach several hundred thousand of points. How your method performs with this amount of data?
Minor comments:
The paper suffers from bad formatting. There are many problems with missing references, equations and symbols floating above the text. There are also some spelling problems.
Line 25: The normal information,
Line 209: via the normal mollification methodology
Line 366: robustness to noise
Line 143, reference for NERL
Algorithms 1, 2, The pseudocode is written with centering, this is not acceptable. Please use left indent and provide the necessary tab spaces to make the code clear.
Overall, the paper presents an interesting approach, but I’m not entirely convinced with the practical application and the comparison with other existing methods. The paper should be improved in these two aspects to obtain a clear understanding of its contribution. Therefore, I recommend a revision.
Reviewer 2 Report
The paper presents a new method for normal estimation in point cloud based on robust identification of sharp features. The method is well presented but have some minor issues to solve. The results show that the method is robust for different classes of point clouds and to the noise. The critical issues are:
1) Improve the presentation of Algorithm 1 and Algorithm 2 using a more readable layout (using coding rules or math rules)
2) In the Related Works, you should add and discuss the following missing references:
- "Estimating surface normals in noisy point cloud data", by Nitra et at.
- "Normal Estimation for Accurate 3D Mesh Reconstruction with Point Cloud Model Incorporating Spatial Structure", by Hashimoto et al.
- "Fast and Accurate Normal Estimation for Point Clouds Via Patch Stitching", by Zhou et al.
- "Deep Learning for Robust Normal Estimation in Unstructured Point Clouds", by Boulch et al.
- "Nesti-Net: Normal Estimation for Unstructured 3D Point Clouds Using Convolutional Neural Networks", by Ben-Shabat et al.
- "Robust Normal Estimation in Unstructured 3D Point Clouds by Selective Normal Space Exploration", by Mura et al.
- "Normal Estimation for 3D Point Clouds via Local Plane Constraint and Multi-scale Selection", by Zhou
Round 2
Reviewer 1 Report
The paper has been improved. However, there are still several formatting errors and some grammar errors (some are denoted below), which in a revised version should not exist.
The experimental section has been improved with comparisons against newer methods. The proposed method has still several parameters that need to be fine-tuned which are subject to noise (like the clustering angle threshold). Also, some parts are not clear, like for instance the RANSAC algorithm against noisy data, which is known to be problematic. It is also not clear if the second stage of the algorithm will eliminate all the problems introduced during the first stage with the “abnormal” points. Based on the claimed contribution I consider this paper at the borderline.
Detail comments:
Line 32 “Next, to account for the practical significance, computational efficiency must be guaranteed.”; Line 45 “we propose a fast and high-quality normal mollification method.”: Given the response to my previous comments please explain how you guarantee computational efficiency of the proposed algorithm, so that the above comments are justified.
Line 131: “However, a lot of time is taken to solve the model in each segmentation process.”: Explain what model has been solved.
Line 132: “To reduce the consumption of computational time”à” To reduce the computational time”
Line 134: “but also ensures its high-quality segmentation”. Why the least-squares method ensures a high-quality segmentation? Does this statement hold for any kind of point clouds?
Algorithm 2 needs better description of the various operations. Provide links to equations and/or section numbers.
Line 284: “1^-4”à”10^-4”
Line 287: Explain what the “outlier normals” are.
Line 292: “or various complications”: such as…?
